# Domain coupling in allosteric regulation of SthK measured using time-resolved transition metal ion FRET

**Pierce Eggan, Sharona E Gordon\*, William N Zagotta\***

Department of Neurobiology and Biophysics, University of Washington, Seattle, United States

## eLife Assessment

This **valuable** study employs transition-metal FRET (tmFRET) and time-correlated single-photon counting to investigate allosteric conformational changes in both isolated cyclic nucleotide-binding domains (CNBDs) and full-length bacterial CNG channels, demonstrating that transmembrane domains stabilize CNBDs in their active state. By comparing isolated CNBD constructs with full-length channels, the authors reveal how allosteric networks couple domain movements to gating energetics, providing insights into ion channel regulation mechanisms. The rigorous methodology and **compelling** quantitative analysis establish a framework for applying tmFRET to study conformational dynamics in diverse protein systems.

**\*For correspondence:**
seg@uw.edu (SEG);
zagotta@uw.edu (WNZ)

**Competing interest:** The authors declare that no competing interests exist.

**Abstract** Cyclic nucleotide-binding domain (CNBD) ion channels are vital for cellular signaling and excitability, with activation regulated by cyclic adenosine- or guanosine-monophosphate (cAMP, cGMP) binding. However, the allosteric mechanisms underlying this activation, particularly the energetics that describe conformational changes within individual domains and between domains, remain unclear. The prokaryotic CNBD channel SthK has been a useful model for better understanding these allosteric mechanisms. Previously, we applied time-resolved transition metal ion Förster resonance energy transfer (tmFRET) to investigate the conformational dynamics and energetics in the CNBD of a soluble C-terminal fragment of the SthK protein, SthK$_{Cterm}$ (Eggan et al., 2024). Here, we used a similar strategy to measure the conformational energetics of the CNBD in the full-length channel, SthK$_{Full}$, and compared them to measurements from SthK$_{Cterm}$. We incorporated the noncanonical amino acid Acd as a FRET donor and a metal bound to a chelator conjugated to a cysteine as an acceptor. We used time-correlated single-photon counting (TCSPC) to measure time-resolved FRET and fit the TCSPC data to obtain donor-acceptor distance distributions in the absence and presence of cAMP. The distance distributions allowed us to quantify the energetics of coupling between the C-terminal domains and the transmembrane domains by comparing the donor-acceptor distance distributions for SthK$_{Cterm}$ and SthK$_{Full}$. Our data indicate that the presence of the SthK transmembrane domains makes the activating conformational change in the CNBD more favorable. These findings highlight the power of time-resolved tmFRET to uncover the structural and energetic landscapes of allosteric proteins and of the ligand-mediated mechanism in CNBD channels specifically.

## Introduction

CNBD ion channels are essential for a variety of physiological processes and are activated when cyclic nucleotides bind to the CNBD of these tetrameric channels (*Craven and Zagotta, 2006*; *He et al.,*

*2014*; *Matulef and Zagotta, 2003*). Upon ligand binding, CNBD channels undergo conformational changes resulting in the opening of the channel pore. However, this pore is over 60 Å away from each of the four peripheral cyclic nucleotide binding sites, and the allosteric mechanisms that underlie this 'activity at a distance' to open the pore remain incompletely understood (*Evans et al., 2020*; *James and Zagotta, 2018*; *Craven et al., 2008*). The prokaryotic CNBD channel SthK has been a useful model to better understand the underlying mechanisms of ligand activation in these channels. Yet, even with substantial structural and functional information of cyclic adenosine monophosphate (cAMP) binding in SthK (*Brams et al., 2014*; *Schmidpeter and Nimigean, 2021*; *Rheinberger et al., 2018*; *Schmidpeter et al., 2018*; *Marchesi et al., 2018*), the mechanism of ligand activation remains unclear from an energetic perspective. As a result, there is a considerable need to integrate both conformational and energetic information in these channels to provide a more complete mechanistic understanding of how cyclic nucleotide binding in the CNBD alters pore opening.

Our previous work utilized time-resolved transition metal ion FRET (tmFRET) to investigate the conformational and energetic changes of the CNBD in an isolated C-terminal fragment of SthK (SthK$_{Cterm}$, *Figure 1A*, left) (*Eggan et al., 2024*). tmFRET uses the efficiency of energy transfer between a donor fluorophore and a light-absorbing metal ion acceptor to measure the molecular distance between the donor and acceptor. Distance distributions were obtained for the movement of the C-helix relative to the beta-roll of the CNBD in the presence and absence of ligand, which allowed for the calculation of the change in free energy (ΔG) and differences in free energy change (ΔΔG) induced by cyclic nucleotide. However, as these SthK$_{Cterm}$ measurements were made in a truncated fragment of the protein, it remained unclear how the CNBD conformational change might look in the full-length channel (SthK$_{Full}$, *Figure 1A*, right) and how the energetics of this transition might differ between the full channel and SthK$_{Cterm}$.

We approach this question using a modular gating scheme, also known as an ensemble allosteric model, which provides a framework for integrating structural and energetic contributions across protein regions, as illustrated in *Figure 1B* (*Horrigan and Aldrich, 2002*; *Motlagh et al., 2014*; *Hilser et al., 2012*). In this model, allosteric transitions occur through coupled conformational changes in defined modules, which for SthK and related CNBD channels, we define as the structural domains of the pore, C-linker, and CNBDs (*Figure 1A*). We depict four independent CNBD transitions and concerted transitions in the C-linker and in the pore, although alternative arrangements and connections are possible (*James and Zagotta, 2018*; *Craven and Zagotta, 2004*; *DeBerg et al., 2016*). Each module has a conformational equilibrium between its resting state (R) and active state (A) (vertical arrows in *Figure 1B*) that is described by a corresponding free energy change, ΔG. Importantly, these transitions in individual modules are not independent but are coupled to each other (horizontal arrows in *Figure 1B*). Coupling is the amount by which the energetics of a module's transition is changed by the state of the neighboring module (ΔΔG). Ultimately, understanding the energetics of all of the arrows in *Figure 1B* is essential to understand the mechanisms of CNBD channel gating.

Quantifying the ΔGs and ΔΔGs for SthK$_{Cterm}$ and SthK$_{Full}$ requires a technique that can measure the conformational changes and energetics of each individual module. Time-resolved tmFRET offers insights into protein conformational and energetic changes by using fluorescence lifetimes to quantify distance distributions between a donor fluorophore and a metal ion acceptor. Importantly, our tmFRET method is applicable to both soluble and membrane protein structures under physiological conditions, making it ideal for studying SthK dynamics and membrane proteins more broadly. The theoretical framework of time-resolved tmFRET has been previously described (*Eggan et al., 2024*; *Haas et al., 1975*; *Zagotta et al., 2024*; *Zagotta et al., 2021*; *Lakowicz, 2006*).

To apply the time-resolved tmFRET approach to full-length SthK channels, we first extended our method to measuring lifetimes in the time domain using time-correlated single photon counting (TCSPC). We believe that TCSPC provides several advantages over our previously utilized frequency-domain measurements on a fluorescence lifetime imaging microscope (FLIM). As membrane protein expression and purification is challenging, we needed an approach that allowed us to use much lower concentrations of protein than required for our previous experiments. The TCSPC cuvette-based instrument has a longer path length than an imaging microscope and, therefore, allowed us to use ~10 fold lower protein concentrations. Additionally, unlike frequency domain measurements, the noise in TCSPC data is due to single-photon shot noise and thus can be well described by a Poisson distribution (above 10 photons), making the $\chi^2$ determinations of model fits more meaningful

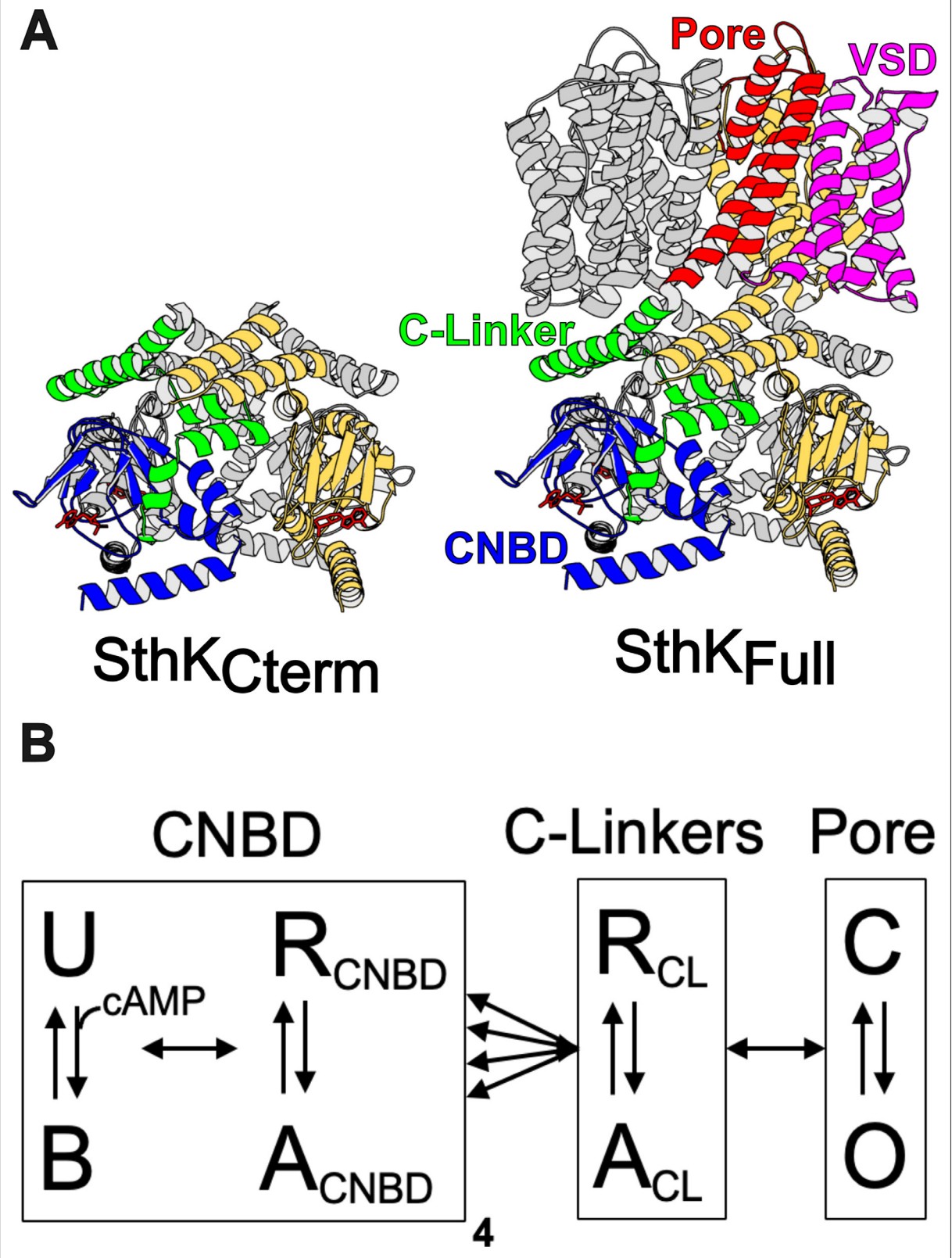

**Figure 1.** Structure of SthK constructs and modular gating scheme. (**A**) Cartoon structures for SthK_Cterm and SthK_Full with the domains labeled on the SthK_Full structure. (**B**) Modular gating scheme for SthK showing modules corresponding to domains indicated in the structures above. Within each domain, resting (R) and active (A) or closed (C) and open (O) or unbound (U) and bound (B) conformations are shown connected by double arrows, representing the transition between states. Horizontal arrows between modules indicate the coupling between domains.

(*Lakowicz, 2006*). Lastly, TCSPC allows for better accounting of the background fluorescence by direct measurement, which allows for more reliably fitting of data regardless of protein concentration or buffer conditions.

As in the frequency domain experiments, lifetimes measured with TCSPC decrease in the presence of FRET. These data can be fit by a model with a distribution of distances that describes the measured FRET between donor and acceptor molecules in the sample. This distance distribution accounts for heterogeneity within and between conformational states of the protein, the latter of which is directly related to the energetics of conformational transitions through the Boltzmann relationship (*Eggan et al., 2024*; *Lakowicz, 2006*).

In this study, we used time-resolved tmFRET with TCSPC data to measure the ligand-induced conformational changes in the CNBD of SthK, first in the isolated C-terminal fragment, $SthK_{Cterm}$, and then in the full-length channel, $SthK_{Full}$. Distance distributions were obtained for both $SthK_{Cterm}$ and $SthK_{Full}$ in the presence and absence of ligand. This allowed for calculation of $\Delta G_{apo}$ and $\Delta G_{cAMP}$ in each construct and the determination of the energetic contribution of coupling to the pore domain. These findings additionally validate the use of TCSPC data in time-resolved tmFRET and demonstrate its capability to measure unique distance distributions across different protein constructs for energetic analysis. Ultimately, these results contribute to a deeper understanding of the allosteric mechanism of ligand gating in SthK.

## Results
### Validation of time-resolved tmFRET using TCSPC in $SthK_{Cterm}$

To first validate our time-resolved tmFRET approach using TCSPC, we started with our previously published construct of the C-terminal fragment of SthK ($SthK_{Cterm}$) (*Eggan et al., 2024*). This allowed us to compare distance distributions obtained with TCSPC with those previously obtained with frequency domain fluorescence lifetime imaging microscopy (FLIM) in the same protein construct. In these experiments, $SthK_{Cterm}$, comprised of the isolated C-linker domain and CNBD, was expressed with the noncanonical amino acid acridon-2-ylalanine, Acd, at S361 and a cysteine at V416 for metal-ion acceptor conjugation. As previously described, $SthK_{Cterm}$ incorporating Acd was purified and mixed with excess WT unlabeled $SthK_{Cterm}$, to create heterotetrameric $SthK_{Cterm}$:WT/S361Acd-V416C protein (cartoon in *Figure 2A*). This heteromeric channel eliminated the possibility of FRET between neighboring subunits (*Eggan et al., 2024*). In parallel, $SthK_{Cterm}$:WT/S361Acd-V416C (with a cysteine) and $SthK_{Cterm}$:WT/S361Acd (donor-only, without any cysteines) protein constructs were incubated with the thiol-specific metal ion acceptor $[Ru(bpy)_2phenM]^{2+}$. This acceptor has an $R_0$ = 43.5 Å when paired with Acd, where $R_0$ is the distance at which FRET efficiency is 0.5 (structures of donor and acceptor labels shown in *Figure 2A*; *Gordon et al., 2024*). The labeled protein was subjected to size exclusion chromatography (SEC) to remove unreacted $[Ru(bpy)_2phenM]^{2+}$ as well as any monomeric protein, and the purified tetrameric protein was used immediately in fluorescence lifetime experiments.

We collected fluorescence lifetime data in the form of TCSPC decays. First, the donor-only $SthK_{Cterm}$:WT/S361Acd was measured and the photon arrival times are shown on a normalized log scale histogram (gray trace, *Figure 2B*). We observed that this lifetime, while approximating a single-exponential decay, was best fit with a double exponential decay, with 87% of the amplitude arising from a 17.6 ns lifetime and the remaining contribution from a second 4.73 ns lifetime. The $SthK_{Cterm}$:S361Acd donor-only protein lifetime did not significantly change in the presence of cAMP, as expected (orange trace, *Figure 2B*). This lifetime is very similar to that previously described for Acd in the same protein using frequency domain lifetime measurements (*Eggan et al., 2024*). These donor-only lifetimes for each condition were then used as fixed parameters when analyzing the amount of FRET in our cysteine-containing protein.

We next measured lifetimes of the $SthK_{Cterm}$:WT/S361Acd-V416C-$[Ru(bpy)_2phenM]^{2+}$ protein, in the absence of ligand (black trace, *Figure 2B*). As expected, labeling this construct with $[Ru(bpy)_2phenM]^{2+}$ decreased the overall lifetime of Acd compared to the donor-only lifetime because of FRET. In addition to decreasing, the lifetime of acceptor-labeled protein became even more non-exponential, indicative of multiple donor-acceptor distances contributing to the FRET. Subsequently, upon the addition of a subsaturating concentration of 1 µM cAMP, the lifetime of Acd decreased (blue trace, *Figure 2B*), and upon addition of a saturating concentration of 320 µM cAMP decreased yet

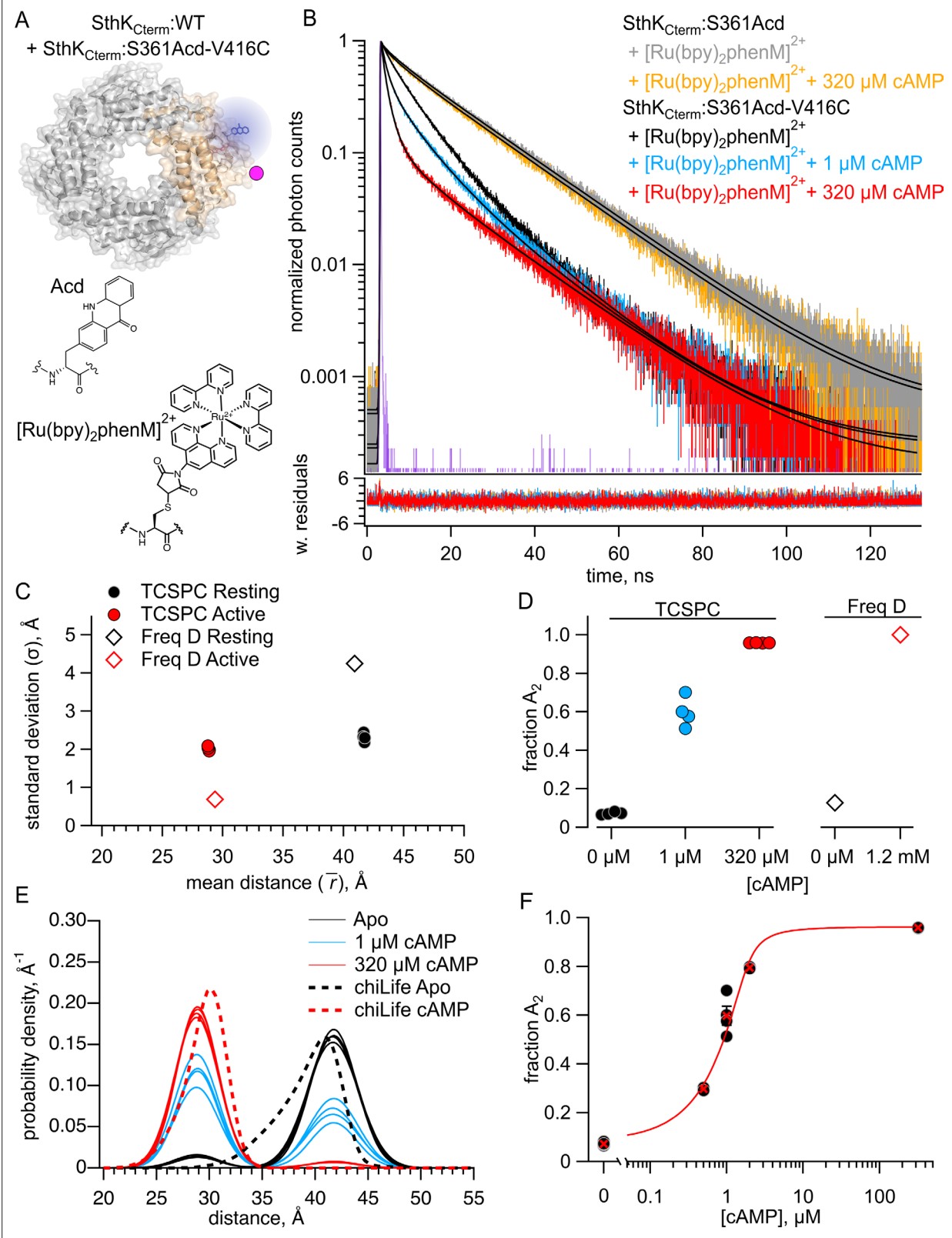

**Figure 2.** Förster resonance energy transfer (FRET) with the donor-acceptor pair within the same subunit. (**A**) Cartoon of SthK$_{Cterm}$:WT/S361Acd-V416C with a single donor-acceptor labeled subunit (top; acceptor shown as magenta dot) and the structures of donor fluorophore, Acd, and acceptor metal complex [Ru(bpy)$_2$phenM]$^{2+}$ (bottom). (**B**) Representative TCSPC data from SthK$_{Cterm}$:WT/S361Acd treated with [Ru(bpy)$_2$phenM]$^{2+}$ before (gray) and after (orange) the addition of 320 µM cAMP. Time-correlated single photon counting (TCSPC) data from SthK$_{Cterm}$:WT/S361Acd-V416C treated with

*Figure 2 continued on next page*

*Figure 2 continued*

[Ru(bpy)$_2$phenM]$^{2+}$ in the absence of cAMP (black) and in the presence of 1 µM (blue) or 320 µM (red) cAMP. Smooth black curves show the fits of the FRET model, and weighted residuals are shown below using the same colors as used for the data. Representative instrument response function (IRF) shown in purple. (**C**) Summary of distance parameter values, $\bar{r}$ and $\sigma$ (n=4), from fits to our TCSPC data (filled symbols) and from previously published frequency domain data (open symbols), with resting and active state as indicated in the legend (*Eggan et al., 2024*). (**D**) $f_{A2}$ values determine from fits to TCSPC data measured with 0, 1 µM and 320 µM cAMP (filled symbols) compared to previously published $f_{A2}$ values determine from frequency domain data (open symbols) (*Eggan et al., 2024*). (**E**) Spaghetti plot of distance distributions from individual experiments for 0 (black), 1 µM (blue) and 230 µM (red) cAMP, with distributions predicted by chiLife shown as dashed curves. Resting and active are shown in black and red, respectively. (**F**) Summary of $f_{A2}$ values over a range of cAMP concentrations. The dose-response relation was fit with a quadratic equation (red curve) with $K_D$ = 0.22 µM and $\left[P\right]_{total}$ = 1.2 µM.

The online version of this article includes the following source data for figure 2:

**Source data 1.** Excel data for time-correlated single photon counting (TCSPC) representative traces, dot plot, scatter plot, spaghetti plot, and dose response plot from Förster resonance energy transfer (FRET) experiments within SthK$_{Cterm}$ subunits (*Figure 2B–F*).

further (red trace, *Figure 2B*). This decrease in the fluorescence lifetime indicates increased FRET and shorter distances between the donor and acceptor molecules in response to cAMP. Overall, these cAMP-induced decreases in lifetimes, compared with our control without acceptor, demonstrate that our TCSPC approach has the sensitivity to measure ligand-triggered conformational changes in the CNBD of SthK$_{Cterm}$.

We fit the SthK$_{Cterm}$:WT/S361Acd-V416C-[Ru(bpy)$_2$phenM]$^{2+}$ lifetimes with a model that assumes a distribution of donor-acceptor distances. We parameterized the distance distribution as the sum of two Gaussians, each with an average distance ($\bar{r}$) and a standard deviation ($\sigma$), as well as fraction $A_2$ ($f_{A2}$) describing the relative contribution of each Gaussian. These two Gaussians represent the resting and active conformational states of the CNBD C-helix. This FRET model was the same that was used for our previously published frequency domain measurements (*Eggan et al., 2024*; *Zagotta et al., 2024*; *Zagotta et al., 2021*; *Lakowicz, 2006*), but adapted to TCSPC lifetime measurements (see Materials and methods).

We globally fit the lifetimes from apo and a range of cAMP concentrations (0.5, 1, 2, 320 µM; *Figure 2B*). These fits converged with a reduced $\chi^2$ near 1, and the parameters were well-identified (see Materials and methods). The $\bar{r}$ and $\sigma$ values for the resting and active state Gaussians from different experiments are summarized in *Figure 2C*, with an average distance $\bar{r}_1$ = 41.7 Å and $\sigma_1$=2.3 Å in the resting state and $\bar{r}_2$ = 28.8 Å and $\sigma_2$=2.1 Å in the active state. These values were similar to our previously published frequency domain results of this same donor-acceptor, however, the $\sigma$ values showed less state dependence (*Figure 2C*; *Eggan et al., 2024*). The fraction of the distribution in the active state, $f_{A2}$, was best fit with 8% in the absence of ligand and 96% in the presence of a saturating concentration of cAMP, again similar to our previous frequency domain measurements (*Figure 2D*). A summary of distributions from individual experiments across various cAMP concentrations is shown in the spaghetti plot in *Figure 2E*. Overlayed on the spaghetti plots are the distance distributions predicted by the known structures using chiLife software (*Eggan et al., 2024*; *Tessmer and Stoll, 2023*). These computationally predicted distance distributions show a remarkable similarity to those based on fits to our experimental data. The $f_{A2}$ values for a range of cAMP concentrations are displayed as a dose-response curve, which was fit with a quadratic equation assuming no binding cooperativity ($K_D$ = 0.22 µM; *Figure 2F*). This $K_D$ value is comparable to the value obtained previously using steady-state tmFRET (*Eggan et al., 2024*). Based on these data, TCSPC determined distance distributions at least as well as our previous frequency domain approach.

## Quantifying intersubunit FRET in SthK$_{Cterm}$

Our ultimate goal of examining conformational energetics in full-length SthK, which includes the membrane spanning domains, presents an additional experimental challenge. Unlike in the C-terminal fragment in which it is straightforward to make heterotetramers including only a single labeled subunit, at this time, we can only express and purify full-length SthK as homotetramers. These homotetramers would, of necessity, have donors and acceptors in all subunits and thus could have intersubunit FRET. To address the contribution of intersubunit FRET, we measured lifetimes in a heterotetramer with Acd incorporated into only a single SthK$_{Cterm}$:S361Acd subunit (with no acceptor cysteine) and [Ru(bpy)$_2$phenM]$^{2+}$ acceptors on the remaining three SthK$_{Cterm}$:V416C subunits (with no Acd donor

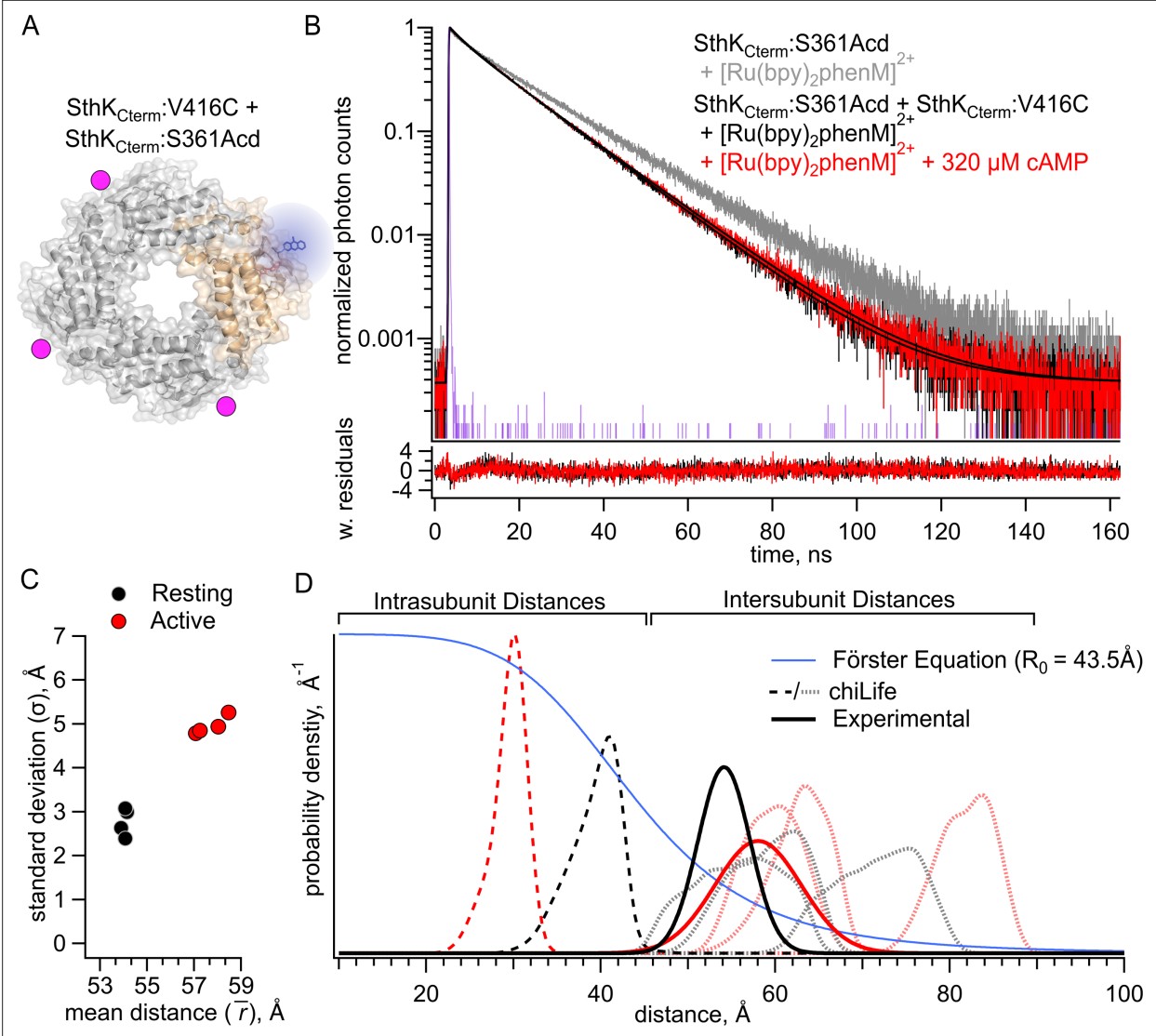

**Figure 3.** Förster resonance energy transfer (FRET) with the donor-acceptor pair on different subunits. (**A**) Cartoon structure of SthK$_{Cterm}$:V416C/S361Acd protein with a single donor-labeled subunit (SthK$_{Cterm}$:S361Acd) and three acceptor-labeled subunits (SthK$_{Cterm}$:V416C). (**B**) Representative TCSPC data from the donor-only condition (gray), and after [Ru(bpy)$_2$phenM]$^{2+}$ labeling either in the absence (black) or presence (red) of 320 µM cAMP, fit with the FRET model (smooth black lines). The instrument response function (IRF) and weighted residuals are depicted as in **Figure 2**. (**C**) Summary of distance parameter values, $\bar{r}$ and $\sigma$ (n=4), from fits of our time-correlated single photon counting (TCSPC) data with the FRET model. Resting and active states are shown as black and red, respectively. (**D**) Distributions of donor-acceptor distances predicted with chiLife for intrasubunit distances (darker dashed curves) and intersubunit distances (lighter dashed curves), with black and red as described in (**C**). The distance distributions from fits to our TCSPC data for resting and active states are overlaid as solid black and red Gaussians.

The online version of this article includes the following source data for figure 3:

**Source data 1.** Excel data for time-correlated single photon counting (TCSPC) representative traces, dot plot, and spaghetti plot from Förster resonance energy transfer (FRET) experiments between SthK$_{Cterm}$ subunits (**Figure 3B–D**).

fluorophores). This protein was produced by mixing an excess of SthK$_{Cterm}$:V416C protein with SthK$_{Cterm}$:S361Acd (**Figure 3A**). We measured shorter lifetimes in these heterotetramers than observed for control donor-only protein, both in the absence and presence of cAMP (**Figure 3B**, black and red traces, respectively). Fits to these data with the FRET model gave parameter values of $\bar{r}_1$ = 54.1 Å and $\sigma_1$ = 3.0 Å in the resting state, and $\bar{r}_1$ = 58.0 Å and $\sigma_1$ = 5.0 Å in the active state (**Figure 3C**). The resting and active distributions aligned well with the intersubunit distance distributions predicted by chiLife (**Figure 3D**). Indeed, the fits most closely aligned with the shortest predicted intersubunit distance, conforming to expectations that the closest distances would dominate the FRET. Based on

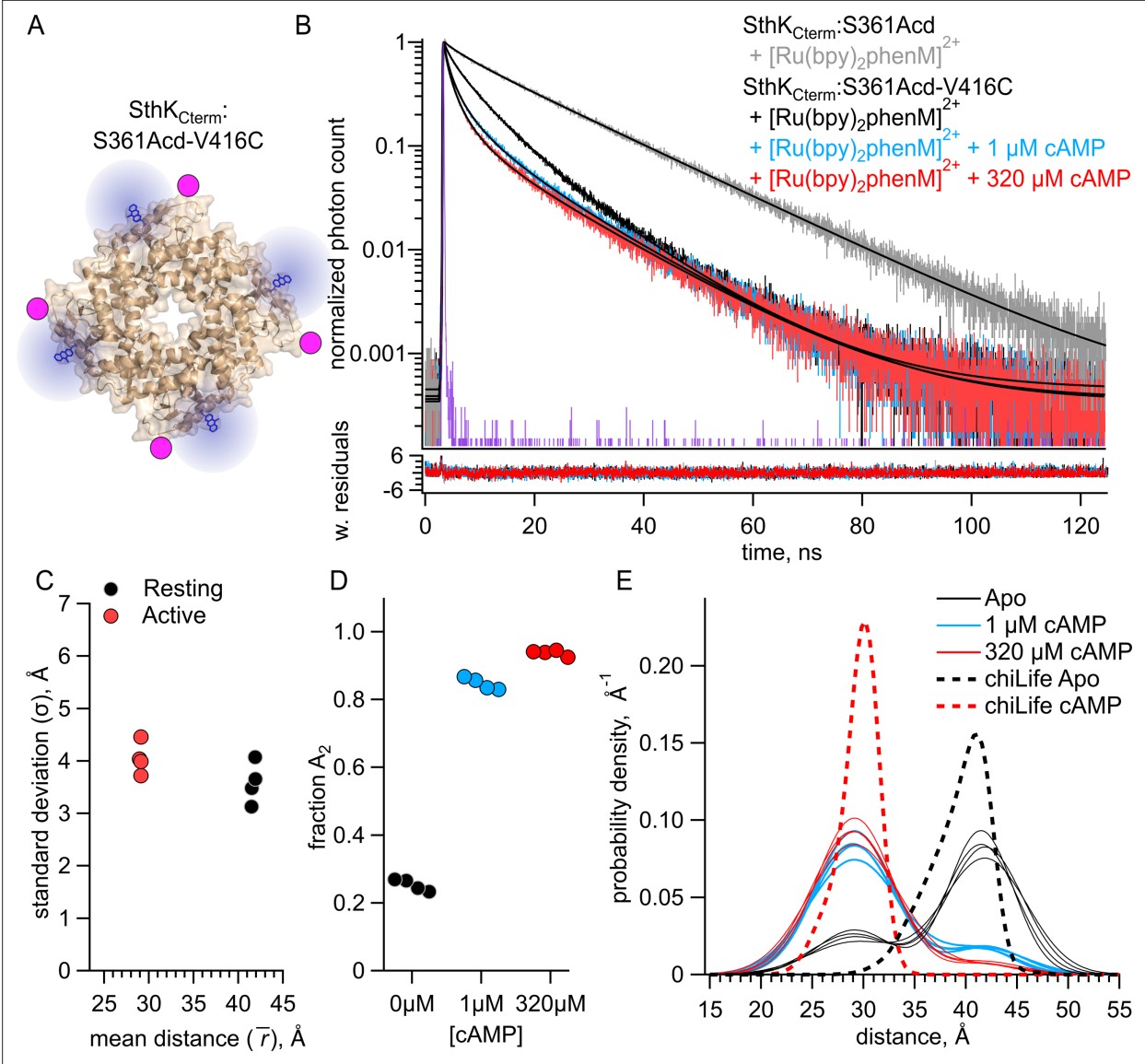

**Figure 4.** Förster resonance energy transfer (FRET) with both intrasubunit and intersubunit donor-acceptor pairs in SthK$_{Cterm}$. (**A**) Cartoon structure of homotetrameric SthK$_{Cterm}$:S361Acd-V416C protein. (**B**) Representative time-correlated single photon counting (TCSPC) data from the donor-only condition (in gray) and after [Ru(Bpy)$_2$phenM]$^{2+}$ labeling in the absence of cAMP (black) and in the presence of either 1 μM (blue) or 320 μM (red) cAMP, fit with the FRET model (smooth black lines). The instrument response function (IRF) and weighted residuals are depicted as in *Figure 2*. (**C**) Summary of distance parameter values, $\bar{r}$ and $\sigma$ (n=4), from fits of our TCSPC data with the FRET model. Resting and active states are shown as black and red, respectively. (**D**) Values of $f_{A2}$ for 0, 1 μM or 320 μM cAMP, as indicated. (**E**) Spaghetti plot of distance distributions from individual experiments for 0 (black), 1 μM (blue) and 230 μM (red) cAMP, with distributions predicted by chiLife shown as dashed curves. Resting and active are indicated by black and red, respectively.

The online version of this article includes the following source data for figure 4:

**Source data 1.** Excel data for time-correlated single photon counting (TCSPC) representative traces, dot plot, scatter plot, and spaghetti plot from experiments with SthK$_{Cterm}$ and inter- and intra-subunit Förster resonance energy transfer (FRET) (*Figure 4B–E*).

these results, we conclude that intersubunit FRET must be considered when fitting data from experiments with homotetrameric SthK with all four subunits labeled with donor and acceptor.

## Measuring homotetrameric SthK$_{Cterm}$ with both intersubunit and intrasubunit FRET

After quantifying the intersubunit FRET in heterotetramers that had FRET only between subunits, we proceeded to measure fluorescence lifetimes in a homotetrameric construct that contained both intrasubunit and intersubunit FRET (*Figure 4A*). We measured lifetimes for homomeric SthK$_{Cterm}$:361Acd-V416C-[Ru(bpy)$_2$phenM]$^{2+}$ in the absence of cAMP and in the presence of either 1 or 320 µM cAMP (*Figure 4B*, black, blue, and red traces, respectively). To account for the contributions from intersubunit FRET, our analysis added a second FRET acceptor with the experimentally determined intersubunit distance distribution (*Figure 3C*, see Materials and methods). Global fits of this FRET model to the data gave intrasubunit distances of $\bar{r}_1$ = 41.6 Å ($\sigma_1$=3.6 Å) and $\bar{r}_2$ = 29.1 Å ($\sigma_2$=4.1 Å), for the resting and active states, respectively (*Figure 4B and C*). These $\bar{r}$ values were similar to those obtained from heteromeric SthK$_{Cterm}$:WT/S361Acd-V416C-[Ru(bpy)$_2$phenM]$^{2+}$, with somewhat wider $\sigma$ values (see *Figure 2C*). Interestingly, the distribution among states in the homotetramer, with a $f_{A2}$ = 0.25 in the absence of ligand, and $f_{A2}$ = 0.94 in the presence of a saturating concentration of ligand, was somewhat different from the SthK$_{Cterm}$:WT/S361Acd-V416C-[Ru(bpy)$_2$phenM]$^{2+}$ heterotetramers (*Figure 4D* vs. *Figure 2D*). This suggests a slightly more favorable closing of the C-helix in the absence of cAMP and slightly less favorable closing in the presence of saturating cAMP in homotetramers. A summary of distance distributions from different experiments across cAMP concentrations is shown as a spaghetti plot in *Figure 4E*, with the chiLife predictions overlayed as dashed curves. Based on these data, we conclude that there are only minimal differences in the distribution among resting and active states between the homotetrameric and heterotetrameric protein.

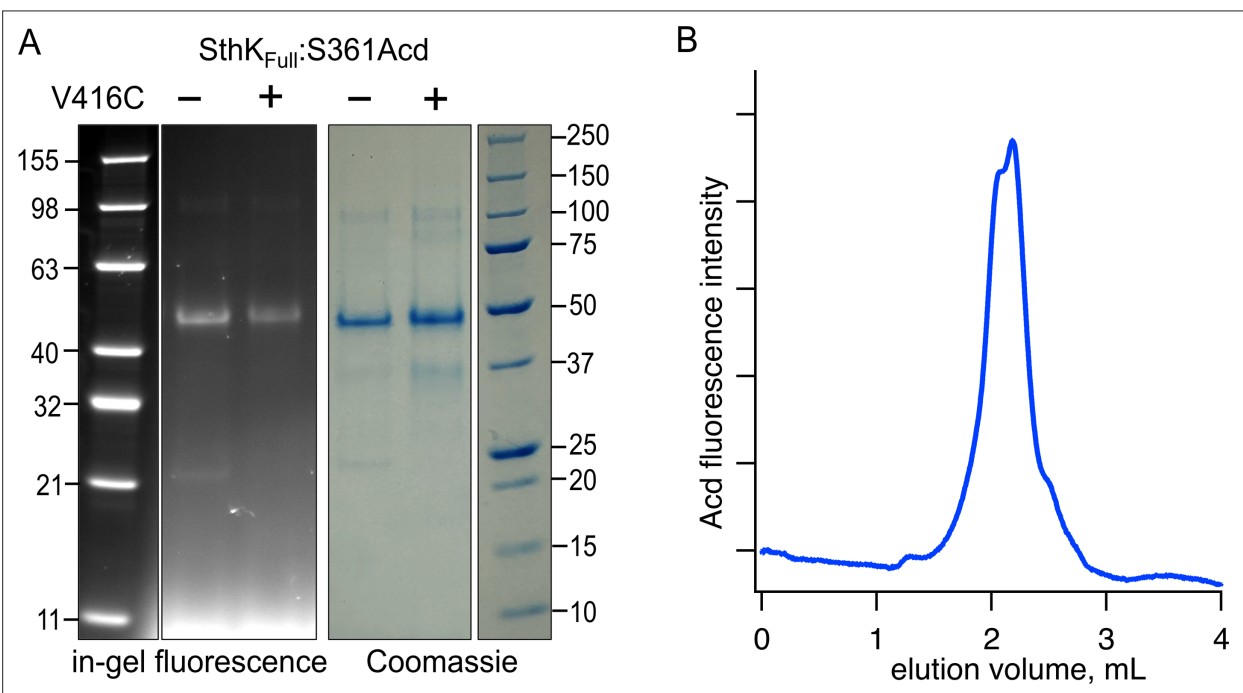

**Figure 5.** Expression and purification of SthK$_{Full}$ incorporating Acd. (**A**) SDS/PAGE with in-gel fluorescence (left) or Coomassie blue (right) for SthK$_{Cterm}$:S361Acd and SthK$_{Cterm}$:S361Acd-V416C, as indicated (theoretical mass = 53.0 kDa). (**B**) Chromatogram from size exclusion chromatography monitoring fluorescence at 425 nm (Acd) for SthK$_{Cterm}$:S361Acd-V416C.

The online version of this article includes the following source data for figure 5:

**Source data 1.** Original uncropped in-gel fluorescence and Coomassie protein gel image (*Figure 5A*).

**Source data 2.** Original labeled in-gel fluorescence and Coomassie protein gels for SthK$_{Full}$ protein (*Figure 5A*).

**Source data 3.** Excel data for size exclusion chromatography (*Figure 5B*).

## Full-length SthK Acd incorporation and purification

Having established a method to account for intersubunit FRET, we expressed and purified full-length SthK protein (SthK$_{Full}$) using the same Acd site at residue 361, either with or without a cysteine at the same acceptor position 416. These constructs were expressed in *E. coli*, where membrane fractions were isolated and solubilized in n-Dodecyl-β-D-maltoside (DDM) detergent with 10:1 cholesteryl hemisuccinate (CHS). The SthK$_{Full}$ was then purified using Strep-Tactin affinity resin, and the DDM detergent was exchanged for Lauryl Maltose Neopentyl Glycol (LMNG) detergent +CHS on the Strep-Tactin column (see Materials and methods). SthK$_{Full}$:S361Acd-V416C (with cysteine) and SthK$_{Full}$:S361Acd (without cysteine) were successfully purified as confirmed with in-gel fluorescence (*Figure 5A*). Coomassie staining showed a small proportion of protein subunits (<15%) that appeared truncated at the TAG codon site and copurified with full-length subunits. Size exclusion chromatography showed predominantly a single peak in Acd fluorescence without aggregates, indicating mono-dispersed well-behaved protein (*Figure 5B*).

## Measuring tmFRET in full-length SthK

We used TCSPC to measure the fluorescence lifetimes of both purified constructs, SthK$_{Full}$:S361Acd-V416C and SthK$_{Full}$:S361Acd, before and after incubating with [Ru(bpy)$_2$phenM]$^{2+}$ acceptor (*Figure 6A*). The lifetime of SthK$_{Full}$:S361Acd donor-only protein was similar to that of SthK$_{Cterm}$:-S361Acd (*Figure 6B*, gray trace), and the addition of cAMP did not change the lifetime (*Figure 6B*, orange trace). As expected, the lifetime of SthK$_{Full}$:S361Acd-V416C became shorter upon labeling with [Ru(bpy)$_2$phenM]$^{2+}$ (*Figure 6B*, black). The addition of a range of cAMP concentrations (0.25, 0.5, 1, 2, and 320 µM) to SthK$_{Full}$:S361Acd-V416C-[Ru(bpy)$_2$phenM]$^{2+}$ protein further decreased the lifetimes, following the same trend as was observed in the SthK$_{Cterm}$ protein (*Figure 6B*).

We globally fit the lifetime data from SthK$_{Full}$ with the FRET model that includes the intrasubunit and intersubunit FRET contributions (*Figure 6B*). We found that the best fit for the dataset had average parameter values of $\bar{r}_1$ = 39.8 Å ($\sigma_1$=4.1 Å) and $\bar{r}_2$ = 31.0 Å ($\sigma_2$=2.7 Å) for the resting and active state Gaussians, respectively (*Figure 6C*). These parameter values are similar to those obtained from homotetrameric SthK$_{Cterm}$, particularly for $\bar{r}s$, which were within 1.5 Å of previous values (*Figure 4C*). The $f_{A2}$ values, however, showed SthK$_{Full}$ had a higher probability of being in the active state, both in the absence of cAMP and in the presence of a saturating concentration of cAMP, compared to SthK$_{Cterm}$ (*Figure 6D*). The distance distributions for the different SthK$_{Full}$ experiments are summarized in spaghetti plots in *Figure 6E*. Across a range of subsaturating cAMP concentrations, $f_{A2}$ showed intermediate values between those obtained from apo and saturating cAMP (*Figure 6F*). This relationship was fit with a quadratic equation to give a K$_D$ = 0.52 µM, which is comparable to that obtained from the SthK$_{Cterm}$ but substantially lower than the K$_D$ reported from WT-SthK electrophysiology experiments (K$_D$ = 1.5 µM) (*Morgan et al., 2019*). These values, however, should be interpreted with caution due to the CNBD's high affinity for cAMP and the challenges of accurately determining the total protein concentrations in our experiments. Overall, in SthK$_{Full}$ containing the pore and transmembrane domains and in a detergent environment, we observed similar resting and active state distances ($\bar{r}$ and $\sigma$) for the CNBD C-helix, but different relative proportions between these states ($f_{A2}$) compared to the SthK$_{Cterm}$.

To focus on changes in $f_{A2}$ in SthK$_{Full}$ compared to SthK$_{Cterm}$, we globally fit the SthK$_{Full}$ and SthK$_{Cterm}$ data assuming the resting and active states average distances ($\bar{r}$) and standard deviations ($\sigma$) were the same across both proteins, and only $f_{A2}$ was allowed to vary. We believe this assumption is reasonable based on the similar distance distributions we found for SthK$_{Full}$ and SthK$_{Cterm}$ (*Figures 6C and 2C*) and the similar structures of the full-length SthK channel in the cAMP-bound and active state (PDB: 7RTJ) and the C-terminal SthK with cAMP bound (PDB: 4D7T) (*Gao et al., 2022*; *Kesters et al., 2015*). Our global fit included data in the absence of cAMP and a saturating concentration of cAMP for both SthK$_{Full}$ and SthK$_{Cterm}$ and used the FRET model with both intersubunit and intrasubunit FRET. The best fit to both datasets yielded $\bar{r}_1$ = 40.5 Å with $\sigma_1$ = 4.1 Å and $\bar{r}_2$ = 30.5 Å with $\sigma_2$ = 3.3 for the absence of cAMP and a saturating concentration of cAMP, respectively. As observed for independent fits to each dataset (*Figures 4D and 6D*), the globally fit $f_{A2}$ differed between the protein constructs, with SthK$_{Full}$ (apo $f_{A2}$ = 0.41, cAMP $f_{A2}$ = 0.99) giving a higher percentage of active conformations in both the absence of cAMP and the presence of a saturating concentration of cAMP, compared to SthK$_{Cterm}$ (apo $f_{A2}$ = 0.12, cAMP $f_{A2}$ = 0.96) (*Figure 6G and H*).

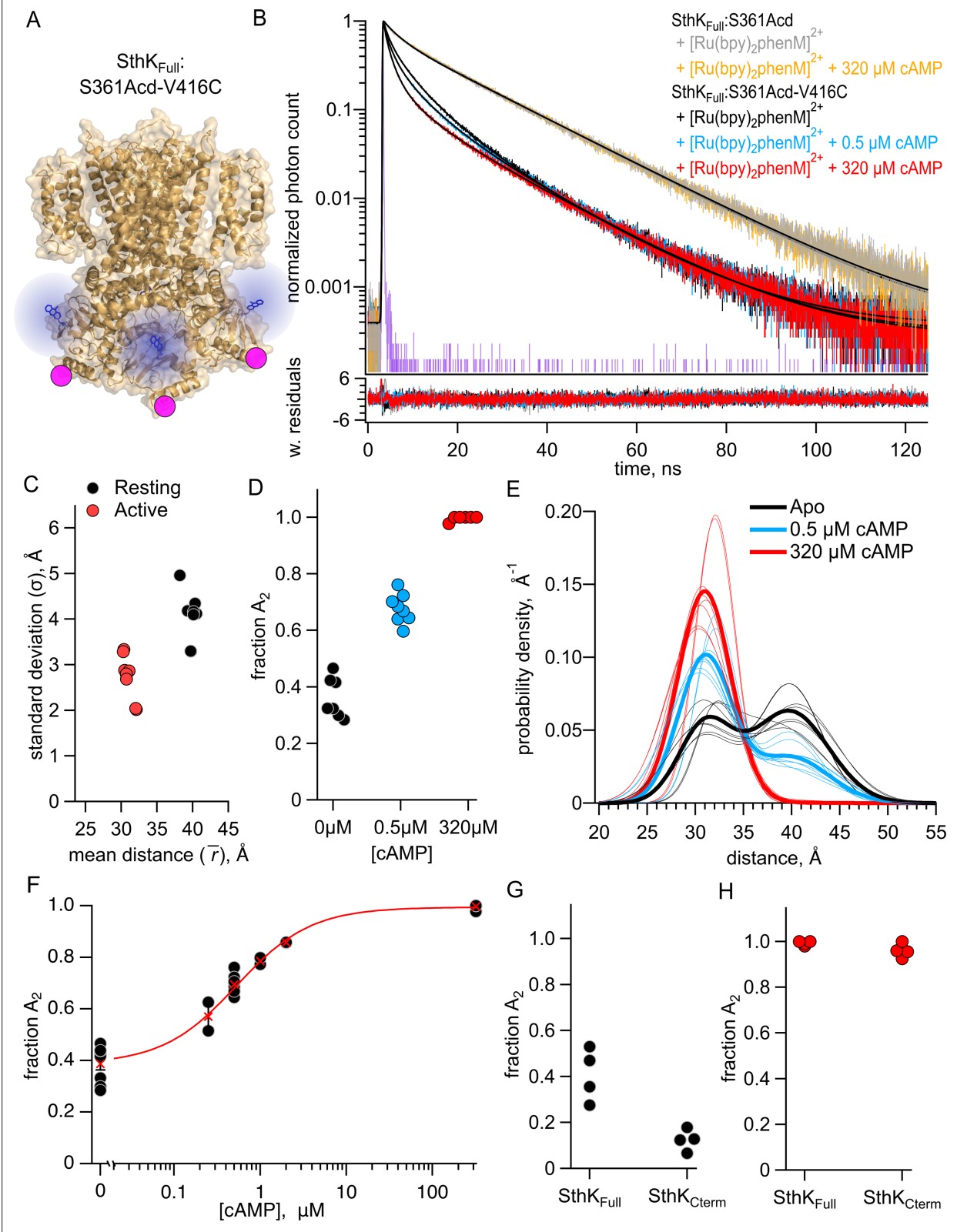

**Figure 6.** Förster resonance energy transfer (FRET) with both intrasubunit and intersubunit donor-acceptor pairs in SthK$_{Full}$. (**A**) Cartoon structure of SthK$_{Full}$:S361Acd-V416C construct. (**B**) Representative time-correlated single photon counting (TCSPC) data from the donor-only condition (no cAMP – gray, 320 μM cAMP – orange) and SthK$_{Full}$:S361Acd-V416C after [Ru(bpy)$_2$phenM]$^{2+}$ labeling in the absence of cAMP (black) and in the presence of either 1 μM (blue) or 320 μM (red) cAMP, fit with the FRET model (smooth black lines). The instrument response function (IRF) and weighted residuals are

*Figure 6 continued on next page*

*Figure 6 continued*

depicted as in **Figure 2**. (**C**) Summary of distance parameter values, $\bar{r}$ and $\sigma$ (n=4), from fits of our TCSPC data with the FRET model. Resting and active states are shown as black and red, respectively. (**D**) Values of $f_{A2}$ for 0, 1 µM or 320 µM cAMP, as indicated. (**E**) Spaghetti plot of distance distributions from individual experiments for 0 (black), 1 µM (blue) and 230 µM (red) cAMP, with distributions predicted by chiLife shown as dashed curves. Resting and active are indicated by black and red, respectively. (**F**) Summary of $f_{A2}$ over range of cAMP concentrations. The dose-response relation was fit with a quadratic equation (red curve) with $K_D$ = 0.53 µM and $[P]_{total}$ = 0.025 µM. (**G–H**) Values of $f_{A2}$ from globally fitting SthK$_{Full}$ and SthK$_{Cterm}$ constructs.

The online version of this article includes the following source data for figure 6:

**Source data 1.** Excel data for time-correlated single photon counting (TCSPC) representative traces, scatter plot, spaghetti plot, dose response plot, and dot plots from experiments with SthK$_{Full}$ (**Figure 2B–H**).

The proportion of molecules in each state can be converted into the change in Gibbs free energy (ΔG) for the CNBD transition in the modular gating scheme in **Figure 1B**. A summary of the energetics for this conformational transition of the C-helix, based on the global fits across the SthK$_{Cterm}$ and SthK$_{Full}$ datasets, is shown in **Figure 7**. In the SthK$_{Cterm}$ protein construct, the resting to active conformational transition is unfavorable in the absence of ligand (ΔG = +1.15 kcal/mol) and is more favorable in the presence of saturating cAMP (ΔG=–1.87 kcal/mol). In comparison, in SthK$_{Full}$ with attached pore and transmembrane domains, the resting to active transition without ligand is still unfavorable (ΔG = +0.22 kcal/mol), but more favorable than in SthK$_{Cterm}$, and cAMP further stabilizes the transition (ΔG=–2.87 kcal/mol). This suggests coupling between the pore and the cytoplasmic domain stabilizes the active state of the CNBD by about –1 kcal/mol. According to the modular gating scheme, domain coupling should not affect the difference in change of free energy (ΔΔG) produced by cAMP, although, of course, it may alter ΔG values in each state. Interestingly, ΔΔG for SthK$_{Cterm}$ (–3.03 kcal/mol) and SthK$_{Full}$ (–3.10 kcal/mol) were very similar. Although the calculated energetics are model

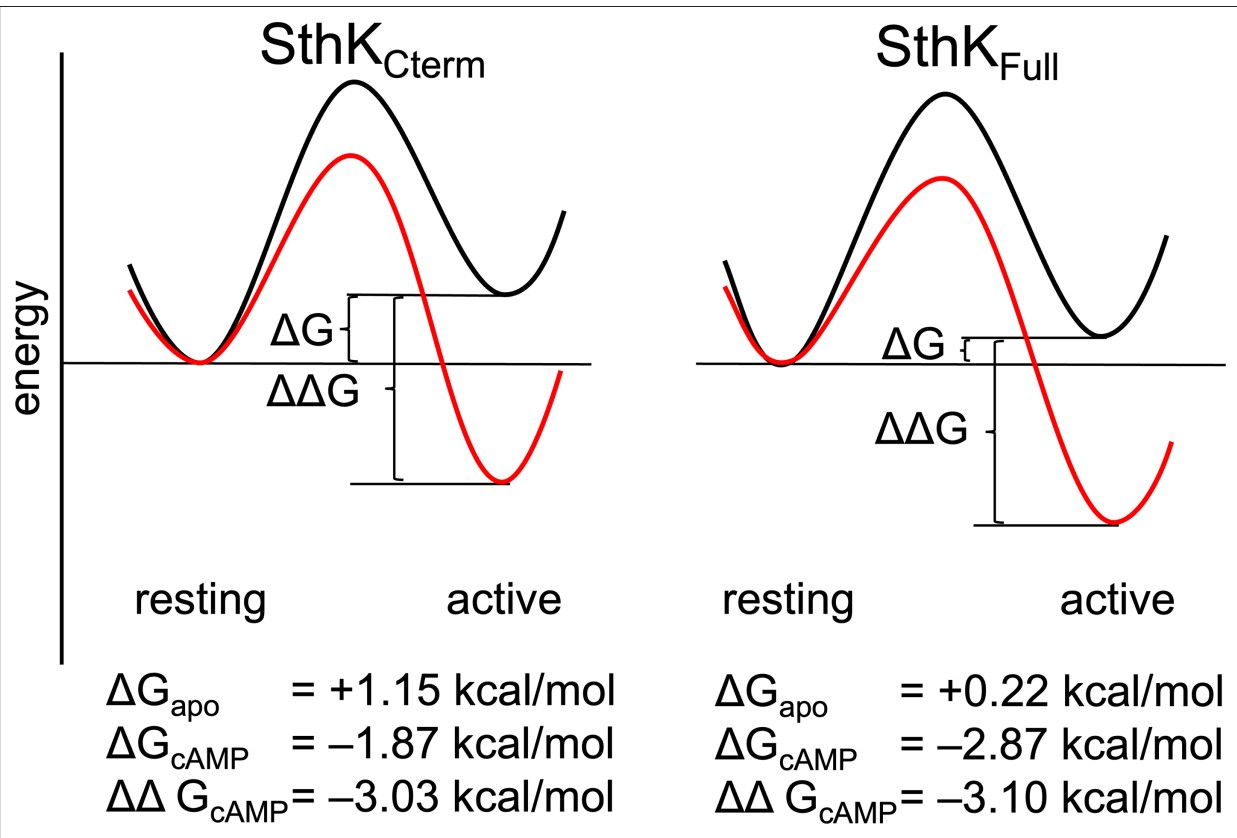

**Figure 7.** Energetics of the resting to active conformational change in the cyclic nucleotide-binding domain (CNBD). A hypothetical reaction coordinate shows the relative energy difference between resting and active states (ΔG) and difference in free energy change (ΔΔG) without ligand (black) and with cAMP bound (red), for SthK$_{Cterm}$ (left) and SthK$_{Full}$ (right). Calculated ΔG and ΔΔGs values are shown for each construct below the diagram.

dependent, the modular gating scheme appears to capture the most salient features of SthK function and provides a context for interpreting our data.

## Discussion

We extended our previous work measuring conformational energetics in the SthK CNBD from a fragment containing the C-terminal C-linker and CNBD to the full-length channel complete with pore and other transmembrane domains. Here, we used the same donor-acceptor pair in the CNBD as in our previous work, albeit with a different method for measuring fluorescence lifetimes. We found that the donor-acceptor distances were remarkably consistent across all constructs, within ~1.5 Å, both in the absence of cAMP and in the presence of a saturating concentration of cAMP. The energetics of the full-length channel (with its pore and transmembrane domain), however, differed from those of the C-terminal fragment. Relative to the C-terminal fragment, the resting-to-activated conformational transition in the full-length channel was stabilized by about −1 kcal/mol, as reflected by a greater $f_{A2}$ value. This stabilization of the active state relative to the resting state in the full-length channel was observed in both the absence of cAMP and in the presence of a saturating concentration of cAMP, i.e., it was not cAMP-dependent. Consequently, the difference in ΔG values between the no ligand and fully-liganded conditions (ΔΔG) was the same for both the C-terminal fragment and full-length channels.

A technical challenge when working with multimeric membrane proteins is the difficulty of producing proteins with only a single labeled subunit. As a result, we measured the potential contributions of FRET between a donor on one subunit of SthK and acceptors on the other three subunits using the C-terminal fragment in which it was straightforward to mix different protein constructs. The small but significant intersubunit FRET we observed required that we consider this contribution in our FRET model with which we fit our lifetime data. After confirming that we could use an adapted FRET model for lifetime data with homotetramers of C-terminal CNBD protein in which all subunits had both donor and acceptor, we applied this method to similarly homotetrameric full-length channels. Although all fits to the data converged and gave $\chi^2$ values near 1, representing intersubunit donor-acceptor distances as arising from a single additional acceptor may not fully capture the intersubunit distance distribution in a tetrameric protein. Nonetheless, this approach improved our tmFRET accuracy for this donor-acceptor site and can be applied to future protein systems where FRET with multiple acceptors cannot be avoided. Another approach would be to use a donor-acceptor pair with a shorter $R_0$, so that the intersubunit FRET is not significant.

There are two extreme, thermodynamically equivalent models for describing the allosteric coupling between the CNBD (including the C-linker) and the pore. The active CNBD could do work on the pore to open it, or the resting CNBD could do work on the pore to close it. These models, both of which are compatible with our modular gating scheme (*Figure 1B*), predict different effects of coupling on the system. In the first model, coupling between the pore and C-terminus makes opening of the pore more favorable and activation of the CNBD less favorable. In the second model, the coupling makes opening of the pore less favorable and activation of the CNBD more favorable. In our experiments, we assumed that deleting the pore and transmembrane domains simply eliminates the coupling of these regions to the CNBD. Our data, showing more favorable activation of the CNBD in the full-length channel containing the pore, are consistent with the second model in which the resting CNBD works on the pore to close it. A similar conclusion was reached for HCN1 channels and spHCN channels, for which deleting the C-terminal domains gave more favorable activation of the pore in electrophysiology experiments (*Vemana et al., 2004*; *Flynn and Zagotta, 2011*; *Wainger et al., 2001*). Our conclusions from the present study are limited because our FRET reporters were only in the CNBD and came from a single donor-acceptor pair, with further advances requiring FRET reporters in the C-linker and pore. Additionally, we did not observe cooperativity between subunits for the CNBD conformational change, and future experiments should investigate where intersubunit cooperativity might arise in these channels. To gain a full mechanistic understanding of allostery in CNBD channels will also require measurements in native lipid bilayers and perhaps even in cells, with their complement of interacting proteins and small molecules.

# Materials and methods

## Key resources table

| Reagent type (species) or resource | Designation | Source or reference | Identifiers | Additional information |
|---|---|---|---|---|
| Strain, strain background (*Escherichia coli*) | B-95.ΔA *E. coli* | Addgene | Bacterial strain #197933 | Electrocompetent cells |
| Recombinant DNA reagent | pDule2-Mj Acd A9 (plasmid) | Addgene | Plasmid #197652 | Provided by GCE4Alls |
| Software, algorithm | TDlifetime_20_Igor_procedures.ipf | This paper | https://github.com/zagotta/TDlifetime_program | Code used for analyzing TCSPC lifetime data for use in tmFRET experiments. |

## Protein expression and purification of SthK$_{Cterm}$

The creation, expression, and purification of SthK$_{Cterm}$ protein constructs was carried out as previously described (***Eggan et al., 2024***). Briefly, all protein constructs were expressed in B-95.ΔA *E. coli* (DE3) cells (***Mukai et al., 2015***), lysed on an Avestin EmulsiFlex-C3 cell disruptor, and lysates were purified on Strep-Tactin high-capacity beads (IBA Biosciences) into KBT (150 mM KCl, 50 mM Tris, 10% glycerol, pH 7.9) with β-mercaptoethanol. For heterotetrameric constructs, an excess of purified subunit (either SthK$_{Cterm}$:WT or SthK$_{Cterm}$:V416C) was added at >7:1 ratio to the purified Acd containing constructs. For SthK$_{Cterm}$:S361Acd-V416C homotetramers, no additional SthK$_{Cterm}$:WT was added. All constructs were TEV cleaved to remove the MBP fusion tag and Twin-strep tag and run on ion exchange chromatography to isolate SthK$_{Cterm}$. Protein was treated with TCEP to ensure cysteines were fully reduced, run on PD MidiTrap G-10 (Cytiva) to remove reducing reagents and exchange buffer back into KBT, and then immediately flash frozen with liquid nitrogen (LN) for storage at –80 °C.

## Constructs, expression, and purification of SthK$_{Full}$

Constructs for the full-length channel SthK$_{Full}$ were created by modifying the previously published construct of the cysteine-free SthK (cfSthK) in the pCGFP vector (***Morgan et al., 2019***). The GFP sequence fragment was removed from the C-terminal end of SthK by Gibson cloning and replaced with the sequences for a TEV protease cleavage sequence followed by a Twin-Strep-tag sequence (the same as the SthK$_{Cterm}$ constructs). Using site-directed mutagenesis, an amber stop codon was introduced at S361 and an acceptor site was engineered by mutating V416 to cysteine to create two constructs: SthK$_{Full}$-S361Acd and SthK$_{Full}$-S361Acd-V416C.

These constructs were co-transformed with the AcdA9 aminoacyl tRNA synthetase/tRNA-containing plasmid (AcdA9.pDule2) (***Sungwienwong et al., 2017***) into B-95.ΔA *E. coli* (DE3) cells (***Mukai et al., 2015***). 1 L cultures of transformed *E. coli* were grown in terrific broth medium at 37 ° C in 100 μg/ml carbenicillin and 120 μg/ml spectinomycin to an OD$_{600}$ ~0.4 before adding Acd (for a final concentration of 0.3 mM) and slowly lowering the temperature to 18 °C (***Speight et al., 2013***; ***Jones et al., 2020***). For protein induction, 0.5 mM isopropyl β-D-1-thiogalactopyranoside (IPTG) was added, and cultures were shaken at 18 °C for 20 hr followed by harvesting by centrifugation. Cell pellets were resuspended in lysis buffer (150 mM KCl, 30 mM Tris, 2 mM β-mercaptoethanol, and 10% glycerol, pH 7.9 supplemented with mini EDTA-free protease inhibitor tablets [Pierce, Thermo Fisher]) and lysed on an Avestin EmulsiFlex-C3 cell disruptor 5 x times at 15,000–20,000 psi. Lysate was diluted and centrifuged at 30,000 x g for 30 min at 4 ° C. Clarified lysate was then spun for 1.25 hr at 200,000 × g at 4 ° C to isolate membranes. *E. coli* membranes were then resuspended 1:2 [wt/vol] in membrane resuspension buffer (150 mM KCl, 30 mM Tris, 20% Glycerol, and 2 mM β-mercaptoethanol, pH 7.9 with an additional mini-protease inhibitor tablet) using a tissue homogenizer. Resuspended membranes were then solubilized with 1:1 [vol/vol] of solubilization buffer (150 mM KCl, 30 mM Tris, 80 mM DDM (Anatrace), 8 mM CHS (Anatrace), 2 mM β-mercaptoethanol, pH 7.9) for 1 hr and added to 1 mL of Strep-Tactin Superflow high-capacity beads (IBA Biosciences) at 4 °C in a disposable column. The resin was washed with 25 mL of 2 mM β-mercaptoethanol supplemented KBT (150 mM KCl, 50 mM Tris, pH 7.9) with 1 mM DDM, 0.1 mM CHS, and then 10 mL of KBT with 4 mM LMNG, 0.4 mM CHS. Detergent was allowed to exchange on the column for 30 mins at 4 °C. An additional 10 mL of KBT was applied with 1 mM LMNG, 0.1 mM CHS with 1 mM tris(2-carboxyethyl)phosphine (TCEP) instead

of β-mercaptoethanol. Protein was eluted from the Strep-Tactin resin with 10 mM d-Desthiobiotin (Sigma) in KBT buffer with 1 mM LMNG and 0.1 mM CHS and 1 mM TCEP.

Detergent-solubilized protein was then run on a bolt 4–12% bis-tris polyacrylamide gel (Invitrogen) for in-gel fluorescence and run analytically on a Superose 6 5/150 size exclusion chromatography (SEC) column (Cytiva) with 0.2 mM LMNG, 0.02 mM CHS to check for aggregates and protein stability. A PD-10 MidiTrap G-10 (Cytiva) desalting column was used to remove TCEP and lower detergent to 0.2 mM LMNG, 0.02 mM CHS. Protein was immediately frozen with LN and stored at –80 °C for later labeling and TCSPC experiments.

## Labeling of SthK$_{Cterm}$ and SthK$_{Full}$ with metal acceptor for tmFRET experiments

All constructs of SthK$_{Cterm}$ (both with cysteine and without) were incubated with 1 mM [Ru(2,2'-bpy)$_2$(1,10-phenanthroline-5-maleimide)]$^{2+}$ ([Ru(bpy)$_2$phenM]$^{2+}$, in DMSO) for 30 min before running on SEC to remove excess label and to isolate tetramers from monomers. Tetrameric SthK$_{Cterm}$ was then used directly in TCSPC experiments.

The SthK$_{Full}$-S361Acd and SthK$_{Full}$-S361Acd-V416C constructs were labeled with 1 mM [Ru(bpy)$_2$phenM]$^{2+}$ for 30 min and then cleaned up as described previously using a BioSpin6 Mini column (BioRad) (*Eggan et al., 2024*).

## TCSPC lifetime measurements

TCSPC lifetime data of Acd-labeled protein samples (85 µL) were measured in 50 µL volume quartz cuvettes (Starna Cells, Inc). Lifetime data were measured using a PicoQuant FluoTime 300 Fluorescence Lifetime Spectrometer (PicoQuant, Berlin, Germany) with 375 nm UV laser excitation and 460 nm emission, and single photon arrivals were recorded on a hybrid photomultiplier detector assembly (PMA-40). Four cuvettes were consecutively recorded, the first two with Acd-labeled protein, the third with buffer-only solution equivalent to the protein sample without protein, and the fourth with diluted Ludox for measurement of the instrument response function (IRF). To each of the first three cuvettes, Adenosine 3',5'-cyclic monophosphate sodium salt monohydrate (cAMP, Sigma Aldrich) in KBT buffer (150 mM KCl, 30 mM Tris, 10% glycerol, pH 7.9) was added at various concentrations following initial measurements without ligand. TCSPC data were acquired with the EasyTau2 measurement software (PicoQuant, Berlin, Germany) and exported for analysis in Igor Pro v8 (Wavemetrics).

## FRET model of time-domain fluorescence lifetime data to obtain Gaussian distance distributions

The time-domain fluorescence lifetime data of the donor fluorophore in the presence of the acceptor were fit by model estimates for the time course of the decay, $Decay(t)$. These fits were calculated from the convolution of the measured instrument response function, $IRF(t)$ of the system with model estimates of the fluorescence lifetime of the donor in the presence of the acceptor, $I_{DA}(t)$ and buffer-only fluorescence $I_B(t)$:

$$Decay(t) = \int_{-\infty}^{\infty} \left[ IRF\left(t' - shift_{irf}\right) - bkgr_{irf} \right] \left[ I_{DA}\left(t - t'\right) + f_B I_B\left(t - t'\right) \right] dt' + bkgr_{dec} \quad (1)$$

where the variables are defined in *Table 1* and displayed graphically in *Figure 8*.

The model for fluorescence lifetime assumes a donor-only fluorescence lifetime with one or two exponential components and FRET between a donor and acceptor separated by one or two Gaussian-distributed distances, similar to that previously described for frequency domain measurements (*Rheinberger et al., 2018*; *Haas et al., 1975*; *Lakowicz, 2006*; *Grinvald et al., 1972*). This model predicts the following relationship for the fluorescence lifetime of the donor in the presence of acceptor, $I_{DA}(t)$:

$$I_{DA}(t) = A_0 \left[ f_D \sum_{i=1}^{2} \alpha_{Di} e^{\left(-\frac{t}{\tau_{Di}}\right)} + (1 - f_D) \int_0^{\infty} \rho(r) \sum_{i=1}^{2} \alpha_{Di} e^{\left[\frac{-t}{\tau_{Di}} - \frac{t}{\tau_{Di}}\left(\frac{R_0}{r}\right)^6\right]} dr \right] \quad (2)$$

**Table 1.** Summary of Variables.

| Variable | Description |
| --- | --- |
| $t$ | Time of photon emission (in s) |
| $r$ | Distance between the donor and acceptor (in Å) |
| $Decay(t)$ | Time-dependent decay of the donor fluorescence (in counts) |
| $DecayB(t)$ | Time-dependent decay of the buffer-only fluorescence (in counts) |
| $I_{DA}(t)$ | Model estimate of the fluorescence lifetime of the donor in the presence of the acceptor (in counts) |
| $I_B(t)$ | Model estimate of the fluorescence lifetime of a buffer-only sample (in counts) |
| $IRF(t)$ | Measured instrument response function (in counts) |
| $bkgr_{dec}$ | Time-independent decay background (in counts) |
| $bkgr_{irf}$ | Time-independent background of the instrument response function (in counts) |
| $shift_{irf}$ | Time shift between the instrument response function and the measured decay (in ps) |
| $f_B$ | Scaling of the buffer fluorescence |
| $A_0$ | Amplitude of the model estimate of the fluorescence lifetime of the donor (in counts) |
| $f_D$ | Fraction of donor-only fluorescence in the sample |
| $\alpha_{Di}$ | Fraction of the ith component of the donor-only fluorescence (in counts) |
| $\tau_{Di}$ | Time constant of the ith component of the donor-only fluorescence (in s) |
| $\alpha_{Bi}$ | Amplitude of the ith component of the buffer fluorescence (in counts) |
| $\tau_{Bi}$ | Time constant of the ith component of the buffer fluorescence (in s) |
| $R_0$ | Characteristic distance between donor and acceptor producing 50% FRET efficiency |
| $\rho(r)$ | Probability distance distribution of the donor and acceptor distances |
| $f_{Ai}$ | Fraction of the ith component of the probability distance distribution |
| $\bar{r}_i$ | Average distance of the ith component of the donor-only fluorescence (in Å) |
| $\sigma_i$ | Standard deviation of the ith component of the donor-only fluorescence (in Å) |

Where $\alpha_{D1} + \alpha_{D2} = 1$.

The density distribution of donor-acceptor distances, $\rho(r)$, was assumed to be the sum of up to two Gaussians:

$$\rho(r) = \sum_{i=1}^{2} f_{Ai} \frac{1}{\sigma_i \sqrt{2\pi}} e^{\left[ \frac{-1}{2} \left( \frac{r - \bar{r}_i}{\sigma_i} \right)^2 \right]} \tag{3}$$

where $f_{A1} + f_{A2} = 1$.

The decay time course of the buffer-only sample, $DecayB(t)$, was fit with a convolution of the measured instrument response function, $IRF(t)$ with a multi-exponential model for the fluorescence lifetime, $I_B(t)$:

$$DecayB(t) = \int_{-\infty}^{\infty} \left[ IRF\left(t' - shift_{irf}\right) - bkgr_{irf} \right] \left[ I_B\left(t - t'\right) \right] dt' + bkgr_{dec} \tag{4}$$

$$I_B(t) = \sum_{i=1}^{4} \alpha_{Bi} e^{\left( \frac{-t}{\tau_{Bi}} \right)} \tag{5}$$

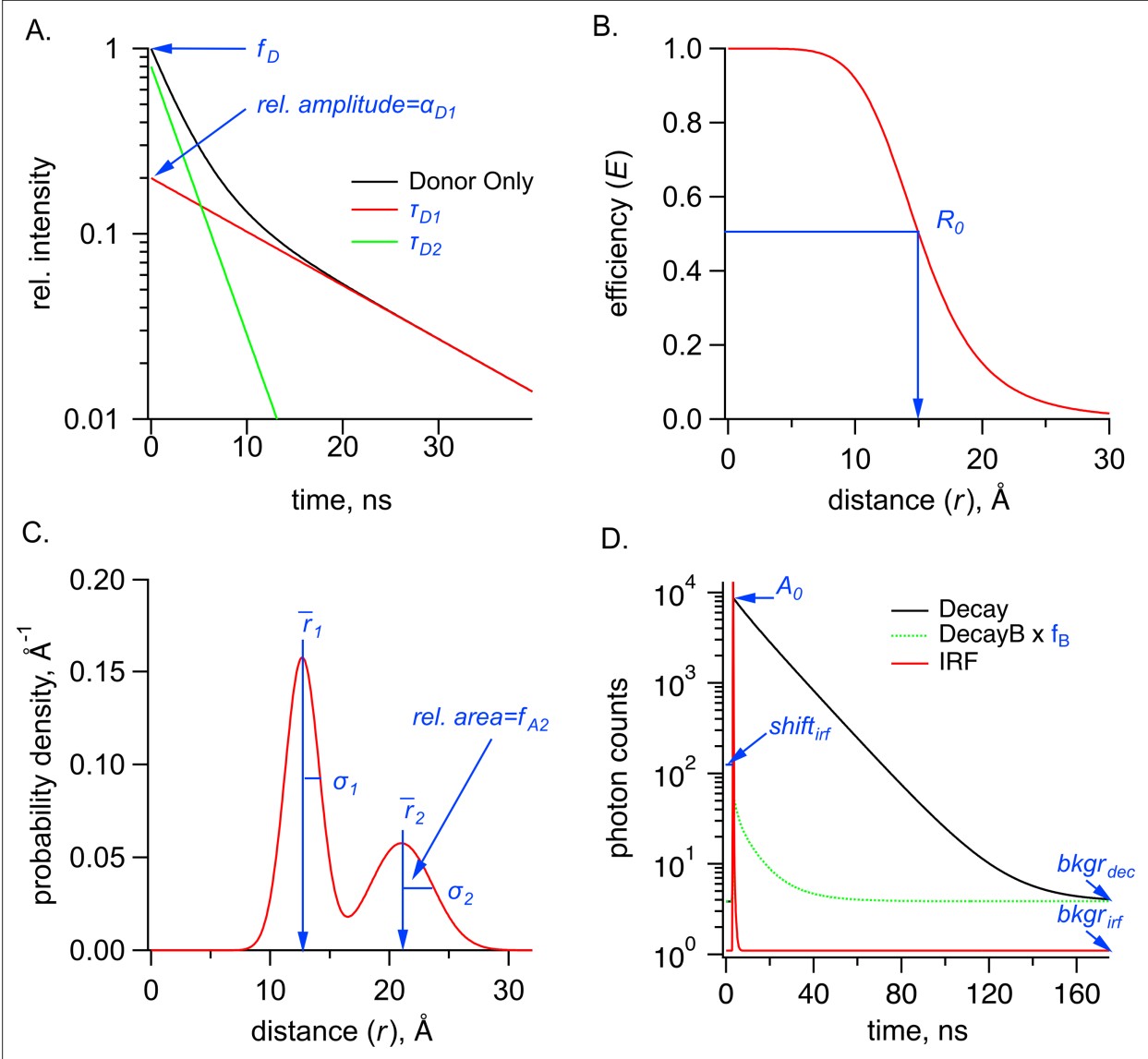

**Figure 8.** Parameters used in the Förster resonance energy transfer (FRET) model, shown in blue, for time-correlated single photon counting (TCSPC) data for the sum of two Gaussian distributions. (**A**) Graph of donor-only fluorescence-lifetime decay with two exponential components with time constants ($\tau_{D1}$, $\tau_{D2}$). The amplitude fraction of the donor-only lifetime in an experiment is determined as $f_D$. (**B**) FRET efficiency (E) plot as a function of distance (r) between donor and acceptor and the $R_0$ values for 50% FRET transfer. (**C**) Probability distribution plot of donor and acceptor distances P(r) showing the sum of two Gaussian distributions, each with their own average distance ($\bar{r}_1$ and $\bar{r}_2$), standard deviations ($\sigma_1$ and $\sigma_2$) and relative amplitude of the second component ($f_{A2}$). (**D**) Example TCSPC data is shown with experimental measured photon count data (Decay), the instrument response function (IRF), and the buffer only decay (DecayB), with its corresponding scaling factor $f_B$ relative to the Decay trace. Also indicated is a time-independent background of photon counts for the Decay trace, $bkgr_{dec}$, as well as two parameters associated with the IRF, its background, $bkgr_{irf}$, and the shift between instrument response function and measured decay, $shift_{irf}$. Parameter $A_0$ is the amplitude of the FRET model estimate of the fluorescence lifetime of the donor (in photon counts).

The online version of this article includes the following source data and figure supplement(s) for figure 8:

**Figure supplement 1.** Identifiability of Individual Model Parameters.

**Figure supplement 1—source data 1.** Excel data of $\chi^2$ minimized curves across various parameters (*Figure 8—figure supplement 1*).

**Figure supplement 2.** Correlation Between Model Parameters.

**Figure supplement 2—source data 1.** Excel data of $\chi^2$ minimized surfaces between various parameters (*Figure 8—figure supplement 2*).

This model for the buffer-only fluorescence lifetime was then added to the model for the sample lifetime before convolving with the $IRF\left(t\right)$ (**Equation 1**).

This FRET model for fluorescence lifetimes and FRET was implemented in Igor Pro v8 (Wavemetrics, Lake Oswego, OR) (code available at https://github.com/zagotta/TDlifetime_program, copy archived at **Zagotta, 2025**). The FRET model was either fit, with $\chi^2$ minimization, assuming a standard deviation equal to $\sqrt{Decay\left(t\right)}$, to individual decay time courses, or globally fit to multiple decays with the sample under different conditions as indicated in the text. There are 15 parameters in the parameter vector $\left(f_D, \tau_{D1}, \alpha_{D1}, \tau_{D2}, R_0, \bar{r}_1, \sigma_1, f_{A2}, \bar{r}_2, \sigma_2, shift_{irf}, f_B, A_0, bkgr_{dec}, bkgr_{irf}\right)$. $R_0$ was fixed to a value previously determined from emission spectra of the donor and absorbance spectra of the acceptor using the Förster equation assuming $\kappa^2 = 2/3$. $f_{A2}$, was generally fixed to 0 or 1 for fits assuming a single Gaussian distance distribution but was allowed to vary when multiple distance components were expected (e.g. with subsaturating ligand concentrations). $f_B$ was fixed to the acquisition time for the experimental sample divided by the acquisition time for the buffer-only sample. And $bkgr_{irf}$ was always fixed to 0 as the average background photon count in the $IRF\left(t\right)$ was <<1.

For the individual decay fits to the donor-only sample, $f_D$ was set to 1, and only $\tau_{D1}, \alpha_{D1}, \tau_{D2}, shift_{irf}, A_0, bkgr_{dec}$ were allowed to vary. For the individual decay fits to the donor +acceptor samples, $\tau_{D1}, \alpha_{D1}, \tau_{D2}$ were constrained to the previously determined values from the donor-only sample, and $f_D, \bar{r}_1, \sigma_1, f_{A2}, \bar{r}_2, \sigma_2, shift_{irf}, A_0, bkgr_{dec}$ were varied. For global fits to multiple decays with the sample under different conditions, multiple parameters were constrained to be the same across some or all of the decay fits, as indicated in the text.

To assess the resolvability of our parameters in our TCSPC FRET model, we generated $\chi^2$ plots for the key parameters (**Figure 8—figure supplement 1**). All parameters we tested reliably converged on a minimized reduced $\chi^2$ value (~1), although the σ parameters produced plots with much shallower slopes and were noticeably less well determined than the other parameters. Additionally, for the parameters expected to have highest correlation, we calculated reduced $\chi^2$ surfaces (**Figure 8—figure supplement 2**). A small correlation was observed between $\sigma_1$ and the fraction donor-only, $f_D$, and between the average distance $\bar{r}_1$ and $f_D$. As a result, we were careful about our interpretation of σ values. Overall, the close agreement of $\bar{r}, \sigma,$ and $f_{A2}$ values obtained from independent measurements and between TCSPC and frequency domain measurements indicate that time-resolved tmFRET is robust across different methods of lifetime measurements.

## Adaptation of the FRET model of time-domain fluorescence lifetime data to include inter-subunit FRET

The relationship for the normalized fluorescence lifetime of the donor when there is a single acceptor is given by a simplified version of **Equation 2**:

$$I_{DA}\left(t\right) = \int_0^\infty \rho\left(r\right) \sum_{i=1}^{2} \alpha_{Di} e^{-\left[\frac{t}{\tau_{Di}} + \frac{t}{\tau_{Di}}\left(\frac{R_0}{r}\right)^6\right]} dr \tag{6}$$

Adding a second FRET acceptor with the same $R_0$ and a new distance $r_2$ gives the following equation for the normalized fluorescence lifetime:

$$I_{DA}\left(t\right) = \int_0^\infty \int_0^\infty \rho_1\left(r_1\right) \rho_2\left(r_2\right) \sum_{i=1}^{2} \alpha_{Di} e^{-\left[\frac{t}{\tau_{Di}} + \frac{t}{\tau_{Di}}\left(\frac{R_0}{r_1}\right)^6 + \frac{t}{\tau_{Di}}\left(\frac{R_0}{r_2}\right)^6\right]} dr_1 dr_2 \tag{7}$$

where $\rho_1\left(r_1\right)$ is the distance distribution of the first acceptor and $\rho_2\left(r_2\right)$ is the distance distribution of the second FRET acceptor. Since the distance distribution of each acceptor is independent of the distance to the other acceptor, this equation can be rewritten as the product of integrals as follows:

$$I_{DA}\left(t\right) = \sum_{i=1}^{2} \int_0^\infty \alpha_{Di} \rho_1\left(r_1\right) e^{-\left[\frac{t}{\tau_{Di}} + \frac{t}{\tau_{Di}}\left(\frac{R_0}{r_1}\right)^6\right]} dr_1 \int_0^\infty \rho_2\left(r_2\right) e^{-\left[\frac{t}{\tau_{Di}}\left(\frac{R_0}{r_2}\right)^6\right]} dr_2 \tag{8}$$

To determine the fluorescence lifetime with both an intrasubunit acceptor and an intersubunit acceptor, the experimentally measured intersubunit FRET (**Figure 3**) was first fit with the lifetime

model from *Equation 2* assuming a single Gaussian distribution for the resting and active states. For simplicity, the FRET distance distribution from three potential intersubunit acceptors was approximated by one Gaussian which largely reflects the closest of the three acceptors. In addition, the fractions of the Gaussian for the activated state ($f_{A2}$) were fixed at values of 0.2 and 0.8 for the apo and cAMP conditions, respectively. The obtained $\bar{r}_1, \sigma_1, \bar{r}_2, \sigma_2$ values from this fit were then used as fixed parameters for the distribution $\rho_2(r_2)$ in *Equation 8* when fitting homotetrameric SthK$_{Cterm}$ and SthK$_{Full}$ lifetime data. While the $\rho_2(r_2)$, $\bar{r}s$ and $\sigma s$ of the intersubunit distance distribution were held constant, the ratio between these Gaussians ($f_{A2}$) was set to the same $f_{A2}$ value as that of $\rho_1(r_1)$. In this way, $\bar{r}_1, \sigma_1, \bar{r}_2, \sigma_2, f_{A2}$ were determined for the intrasubunit distance distribution $\rho_1(r_1)$.

## Free energy and dose-response calculations

Free energies were calculated using the following equations:

$$\Delta G = -RT * ln\left(\frac{f_{A2}}{1 - f_{A2}}\right) \tag{9}$$

$$\Delta\Delta G_{cAMP} = \Delta G_{cAMP} - \Delta G_{apo} \tag{10}$$

where $R$ is the universal gas constant, $T$ is the absolute temperature (K), and $f_{A2}$ and $1 - f_{A2}$ are the proportion of molecules in the active and resting states respectively.

Dose-response relations (*Figures 2F and 6F*) were fit by the following quadratic equation:

$$Fractional\ Response = (A - B) \times \frac{\left([L]_{total} + [P]_{total} + K_D\right) - \sqrt{\left([L]_{total} + [P]_{total} + K_D\right)^2 - 4 \times [P]_{total} \times [L]_{total}}}{2 \times [P]_{total}} + B \tag{11}$$

Where $K_D$ is the dissociation constant for cAMP, $[L]_{total}$ is the total ligand (cAMP) concentration, $[P]_{total}$ is the total protein concentration, $A$ is the $f_{A2}$ value at saturating ligand concentrations and $B$ is the $f_{A2}$ value without ligand. The above parameters were obtained from least squares fits to the data, and the protein concentration $[P]_{total}$ was corroborated via A$_{280}$ and Bradford assay for each sample.

## chiLife predictions

Computational predictions of the possible rotameric positions for the donor and acceptor labels were made with chiLife using the accessible-volume sampling method as previously described (*Zagotta et al., 2024*; *Tessmer and Stoll, 2023*). Acd and [Ru(bpy)$_2$phenM]$^{2+}$ were added as custom labels in chiLife and modeled onto the cryo-EM structure of the full-length SthK closed state (*Gao et al., 2022*) (PDB: 7RSH, residues 225–416) and the X-ray crystallography structure of the cAMP-bound SthK C-terminal fragment (*Kesters et al., 2015*) (PDB: 4D7T). For each donor-acceptor pair, labels were superimposed at indicated residue positions, and 10,000 possible rotamers were modeled. Rotamers resulting in internal clashes (<2 Å) were removed and external clashes evaluated as previously described (*Tessmer and Stoll, 2023*). Donor-acceptor distance distributions were calculated between the remaining (~500–2000) label rotamers for each donor-acceptor pair. Inter-subunit distances were calculated as distances between one Acd molecule and the modeled acceptor rotamers on each of the other three subunits. Structural representations were made using PyMol (V3.0, Schrödinger, LLC).

## Acknowledgements

We thank the Oregon State University GCE4ALL (Center for Genetic Code Expansion for All) for their long-standing collaboration, and Drs. James Petersson, and Kyle D Shaffer (University of Pennsylvania) for excellent technical support with Acd. Thanks to Drs. Max Tessmer and Stefan Stoll (University of Washington) for technical assistance with chiLife. We also thank all members of the SEG and WNZ Laboratories for helpful conversations and support. Research reported in this publication was supported by the National Institutes of Health under award numbers R35GM145225 (to SEG), R35GM148137 and R01EY010329 (to WNZ), T32GM008268 and T32EY007031 (to PE).

# Additional information

## Funding

| Funder | Grant reference number | Author |
|---|---|---|
| National Institute of General Medical Sciences | R35GM148137 | William N Zagotta |
| National Eye Institute | R01EY010329 | William N Zagotta |
| National Institute of General Medical Sciences | R35GM145225 | Sharona E Gordon |
| National Institute of General Medical Sciences | T32GM008268 | Pierce Eggan |
| National Eye Institute | T32EY007031 | Pierce Eggan |

The funders had no role in study design, data collection and interpretation, or the decision to submit the work for publication.

## Author contributions

Pierce Eggan, Conceptualization, Data curation, Formal analysis, Investigation, Visualization, Methodology, Writing - original draft, Writing – review and editing; Sharona E Gordon, Conceptualization, Data curation, Formal analysis, Funding acquisition, Investigation, Methodology, Writing – review and editing; William N Zagotta, Conceptualization, Data curation, Software, Formal analysis, Funding acquisition, Investigation, Methodology, Writing – review and editing

## Author ORCIDs

Pierce Eggan ⓘ https://orcid.org/0000-0001-6134-4569
Sharona E Gordon ⓘ https://orcid.org/0000-0002-0914-3361
William N Zagotta ⓘ https://orcid.org/0000-0002-7631-8168

Reviewer #1 (Public review): https://doi.org/10.7554/eLife.106892.3.sa1
Reviewer #2 (Public review): https://doi.org/10.7554/eLife.106892.3.sa2
Reviewer #3 (Public review): https://doi.org/10.7554/eLife.106892.3.sa3
Author response https://doi.org/10.7554/eLife.106892.3.sa4

# Additional files

## Supplementary files

MDAR checklist

## Data availability

All data generated or analyzed during this study are included in the manuscript and supporting files. Code used for data analysis has been posted to https://github.com/zagotta/TDlifetime_program (copy archived at *Zagotta, 2025*).

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
