## [Editor Report · eLife Assessment]

This **valuable** study employs transition-metal FRET (tmFRET) and time-correlated single-photon counting to investigate allosteric conformational changes in both isolated cyclic nucleotide-binding domains (CNBDs) and full-length bacterial CNG channels, demonstrating that transmembrane domains stabilize CNBDs in their active state. By comparing isolated CNBD constructs with full-length channels, the authors reveal how allosteric networks couple domain movements to gating energetics, providing insights into ion channel regulation mechanisms. The rigorous methodology and **compelling** quantitative analysis establish a framework for applying tmFRET to study conformational dynamics in diverse protein systems.

---

## [Referee Report · Reviewer #1 (Public review)]

Summary:

This useful work extends a prior study from the authors to observe distance changes within the CNBD domains of a full length CNG channel based on changes in single photon lifetimes due to tmFRET between a metal at an introduced chelator site and a fluorescent non canonical amino acid at another site. The data are excellent and convincingly support the authors' conclusions. In addition to the methodology being of general use for other proteins, the authors show that coupling of the CNBDs to the rest of the channel stabilizes the CNBDs in their active state relative to an isolated CNBD construct.

Strengths:

The manuscript is very well written and clear.

---

## [Referee Report · Reviewer #2 (Public review)]

The manuscript by Eggan et al. investigates the energetics of conformational transitions in the cyclic nucleotide-gated (CNG) channel SthK. This lab pioneered transition metal FRET (tmFRET), which has previously provided detailed insights into ion channel conformational changes. Here, the authors analyze tmFRET fluorescence lifetime measurements in the time domain, yielding detailed insights into conformational transitions within the cyclic nucleotide binding domains (CNBDs) of the channel. The integration of tmFRET with time-correlated single-photon counting (TCSPC) represents an advancement of this technique.

---

## [Referee Report · Reviewer #3 (Public review)]

Summary:

This is a lucidly written manuscript describing the use of transition-metal FRET to assess distance changes during functional conformational changes in a CNG channel. The experiments were performed on an isolated C-terminal nucleotide binding domain (CNBD) and on a purified full-length channel, with FRET partners placed at two positions in the CNBD.

The data and quantitative analysis are exemplary, and they provide a roadmap for the use of this powerful approach in other proteins. In particular, the use of the fluorescence-lifetime decay histograms to learn not just the mean distance reported by the FRET, but also the distribution of states with different distances, allows better refinement of hypotheses for the gating motions.

---

## [Author Response]

The following is the authors’ response to the original reviews

**Public Reviews:**

**Reviewer #1 (Public review):**
Summary:This useful work extends a prior study from the authors to observe distance changes within the CNBD domains of a full-length CNG channel based on changes in single photon lifetimes due to tmFRET between a metal at an introduced chelator site and a fluorescent non-canonical amino acid at another site. The data are excellent and convincingly support the authors' conclusions. The methodology is of general use for other proteins. The authors also show that coupling of the CNBDs to the rest of the channel stabilizes the CNBDs in their active state, relative to an isolated CNBD construct.Strengths:The manuscript is very well written and clear.
**Reviewer #2 (Public review):**
The manuscript "Domain Coupling in Allosteric Regulation of SthK Measured Using Time-Resolved Transition Metal Ion FRET" by Eggan et al. investigates the energetics of conformational transitions in the cyclic nucleotide-gated (CNG) channel SthK. This lab pioneered transition metal FRET (tmFRET), which has previously provided detailed insights into ion channel conformational changes. Here, the authors analyze tmFRET fluorescence lifetime measurements in the time domain, yielding detailed insights into conformational transitions within the cyclic nucleotide binding domains (CNBDs) of the channel. The integration of tmFRET with time-correlated single-photon counting (TCSPC) represents an advancement of this technique.The results summarize known conformational transitions of the C-helix and provide distance distributions that agree with predicted values based on available structures. The authors first validated their TCSPC approach using the isolated CNBD construct previously employed for similar experiments. They then study the more complex fulllength SthK channel protein. The findings agree with earlier results from this group, demonstrating that the C-helix is more mobile in the closed state than static structures reflect. Upon adding the activating ligand cAMP, the C-helix moves closer to the bound ligand, as indicated by a reduced fluorescence lifetime, suggesting a shorter distance between the donor and acceptor. The observed effects depend on the cAMP concentration, with affinities comparable to functional measurements. Interestingly, a substantial amount of CNBDs appear to be in the activated state even in the absence of cAMP (Figure 6E and F, fA2 ~ 0.4).This may be attributed to cooperativity among the CNBDs, which the authors could elaborate on further. In this context, the major limitation of this study is that distance distributions are observed only in one domain. While inter-subunit FRET is detected and accounted for, the results focus exclusively on movements within one domain. Thus, the resulting energetic considerations must be assessed with caution. In the absence of the activator, the closed state is favored, while the presence of cAMP favors the open state. This quantifies the standard assumption; otherwise, an activator would not effectively activate the channel. However, the numerical values of approximately 3 kcal/mol are limited by the fact that only one domain is observed in the experiment, and only one distance (C- helix relative to the CNBD) is probed. Additional conformational changes leading to pore opening (including rotation and upward movement of the CNBD, and radial dilation of the tetrameric assembly) are not captured by the current experiments. These limitations should be taken into account when interpreting the results.

We agree that these are important limitations to consider in interpreting our results. These limitations and future directions are now largely covered in our discussion. We believe measurements in individual domains provide unique insights into the contributions of different parts of the protein and future work will continue to address conformational energetics in other parts of the protein and subunit cooperativity.

**Reviewer #3 (Public review):**
Summary:This is a lucidly written manuscript describing the use of transition-metal FRET to assess distance changes during functional conformational changes in a CNG channel.The experiments were performed on an isolated C-terminal nucleotide binding domain(CNBD) and on a purified full-length channel, with FRET partners placed at twopositions in the CNBD.Strengths:The data and quantitative analysis are exemplary, and they provide a roadmap for use of this powerful approach in other proteins.Weaknesses/Comments:A ~3x lower Kd for nucleotide is seen for the detergent-solubilized full-length channel, compared to electrophysiological experiments. This is worth a comment in the Discussion, particularly in the context of the effect of the pore domain on the CNBD energetics.

We are cautious to interpret our K_D_ values given the high affinity for cAMP and the challenges of accurately determining the total protein concentrations in our experiments. We now state this explicitly in the manuscript.

**Recommendations for the authors:**

**Reviewer #1 (Recommendations for the authors):**
The manuscript is very well written and clear. Congrats to the authors.Minor comment: In "Measuring tmFRET in Full-Length SthK", 3rd paragraph: "... FRET model with both intersubunit and intersubunit FRET." Should read "intersubunit and intrasubunit".

Thank you for the comment, this is now corrected.

**Reviewer #2 (Recommendations for the authors):**
Overall, the manuscript is well-written and clearly explained. However, I recommend that the authors discuss the limitations more critically.

The revised manuscript now largely addresses these limitations. Additional comments are addressed in short below:

A) Only one distance is measured.

We believe validating a single distance as an important first step in determining the use of this technique and beginning to quantify the allosteric mechanism in SthK. Future studies aim to make additional measurements.

B) Measurements are confined to a single domain in the cooperative tetrameric assembly.

Isolating conformational changes in individual domains, allows us to determine how different parts of the protein contribute to the activation upon ligand binding.

C) The change in distance upon activation mirrors what is observed in the closed state, which casts doubt on whether these conformational changes actually lead to channel opening or merely reflect the upward swinging of the C-helix that contributes to coordinating cAMP in the binding pocket.

Future studies aim to detect conformational changes in the pore and other parts of the protein.

D) Rigid body movements, rotations, and dilations are not captured by the measurements.

Our measurements combine energetic information with some, although more limited, structural information.

E) Cooperativity is not considered in the interpretation of the results.

It is currently unclear where in SthK cooperativity arises upon ligand activation (ie. at the level of the CNBD, C-Linker or pore). Our results do not provide evidence of cooperativity in the CNBD upon ligand binding.

Additionally, the authors directly correlate their results with the functional states of SthK previously reported, but it remains open whether the modified protein for tmFRET behaves similarly to WT SthK. Functional experiments with the protein used for tmFRET, which demonstrate comparable open probabilities and cAMP potency, would considerably strengthen the manuscript.

Further optimization is needed to express the full-length protein used in tmFRET experiments in spheroplasts to enable electrophysiological recordings from these constructs.

**Reviewer #3 (Recommendations for the authors):**
In the final paragraph of the Discussion, the sentence "In our experiments, we assumed that deleting the pore and transmembrane domains eliminates the coupling of these regions to the CNBD" seems trivial. Perhaps it would help to add "simply" before eliminates?

We have taken the advice and added ‘simply’ in this sentence.

Can a statement be made about the magnitude of the effect in the C-terminal deletion experiments in refs 27-29?

Due to the different channels used in the C-terminal deletion experiments in refs 27-29 (HCN1 and spHCN), compared to the channel we used (SthK), it is challenging to compare the magnitude of energetic changes between these studies. Additionally, the HCN experiments measured changes in the pore domain, compared to the conformational changes in the CNBD domain measured here.